# The Vacuolar Morphogenesis Protein Vam6-Like Protein Vlp1 Is Required for Pathogenicity of *Cryptococcus neoformans*

**DOI:** 10.3390/jof7060418

**Published:** 2021-05-27

**Authors:** Cheng-Li Fan, Tong-Bao Liu

**Affiliations:** 1College of Animal Science and Technology, Southwest University, Chongqing 400715, China; victoriafan@swu.edu.cn; 2State Key Laboratory of Silkworm Genome Biology, Southwest University, Chongqing 400715, China; 3Medical Research Institute, Southwest University, Chongqing 400715, China

**Keywords:** *Cryptococcus neoformans*, vacuolar morphogenesis protein, Vlp1, virulence, macrophage

## Abstract

*Cryptococcus neoformans* is an encapsulated yeast pathogen that infects immunocompromised patients to cause fungal meningitis, resulting in hundreds of thousands of deaths each year. F-box protein Fbp1, the key component of the E3 ubiquitin ligase, plays a critical role in fungal development and virulence in fungal pathogens. In this study, we identified a potential substrate of Fbp1, the vacuolar morphogenesis protein Vam6-like protein Vlp1, and evaluated its role in virulence in *C. neoformans*. Deletion or overexpression of the *VLP1* gene results in abnormal capsule formation and melanin production of *C. neoformans*. Stress tolerance assay showed that the *vlp1*Δ mutant was sensitive to SDS and NaCl but not to CFW or Congo red, indicating that Vlp1 might regulate the cell membrane integrity in *C. neoformans*. Fungal virulence assay showed that Vlp1 was essential for the pathogenicity of *C. neoformans*, as *vlp1*Δ mutants are avirulent in the mouse systematic infection model of cryptococcosis. The progression of fungal infection revealed that the *vlp1*Δ mutants were gradually eliminated from the lungs of the mice after infection. Moreover, the *vlp1*Δ mutants showed a proliferation defect inside macrophages and a viability defect in the host complement system, which likely contributes to the virulence attenuation of the *vlp1*Δ mutants. In summary, our results revealed that the vacuolar morphogenesis protein Vam6-like protein Vlp1 is essential for the pathogenicity of *C. neoformans*.

## 1. Introduction

*Cryptococcus neoformans* is an encapsulated yeast-like pathogen that exists widely in nature. The basidiospores and desiccative yeast cells of *C. neoformans* can enter the human lungs through breathing, causing cryptococcal pneumonia and meningitis [1]. In recent years, with the increase in immunodeficient or immunocompromised patients, the morbidity and mortality caused by *C. neoformans* are also on the rise [2,3]. According to the latest estimates, *C. neoformans* causes at least 278,000 infections each year and leads to at least 181,000 deaths [4]. If the infection is not treated in time, the mortality rate is 100% [5]. Thus far, cryptococcal infection has become one of the common complications of AIDS patients and is responsible for 15% of AIDS-related deaths globally [4]. Thus, although the pathogenesis of *C. neoformans* has been studied extensively in the last few decades, it is still not very clear and attracts global attention.

The ubiquitin–proteasome system (UPS) is the main protein degradation system in cells, composed of ubiquitin-activating enzyme E1, ubiquitin-conjugating enzyme E2, and ubiquitin ligase E3, and is responsible for the degradation of more than 80% of intracellular proteins [6,7]. Among the numerous E3 ligases, the Skp1, Cullin, F-box protein (SCF) E3 ligases are the most prominent E3 gene family, and F-box protein is the key component that determines the specificity of the substrate [8]. Thus far, many F-box proteins have been identified in fungi, and gene function analysis shows that F-box proteins play an important role in regulating transcription, cell cycle, circadian clocks, development, signal transduction, and virulence [9,10]. Previously, we identified an F-box protein Fbp1 in *C**. neoformans*, and functional analysis showed that the deletion of Fbp1 resulted in the complete loss of pathogenicity of *C**. neoformans* [11]. Further mechanism investigation of Fbp1 function during infection showed that the *fbp1*Δ mutant had an intracellular proliferation defect after phagocytosis in a *Cryptococcus*–macrophage interaction assay [12]. Our subsequent studies have shown that the hypovirulence of the *fbp1*Δ mutant is linked to the development of a robust host immune response [13]. However, as an F-box protein, the downstream substrates of cryptococcal Fbp1 and how it affects the morphological development and pathogenicity of *C*. *neoformans* by regulating the downstream are still unknown.

In our previous study, an iTRAQ-based proteomics approach was used to identify the downstream substrates of Fbp1 in *C**. neoformans* [14]. Among the candidate proteins, the vacuolar morphogenesis protein Vam6-like protein Vlp1 (CNAG_05395) was found to have three PEST domains and could be a potential substrate of Fbp1 and is worth investigating. In this study, we set out to further identify and investigate the function of Vlp1. We constructed the *VLP1* gene deletion, complemented, and overexpressed strains using a split marker strategy and biolistic transformation method and tested the formation of virulence factors of the strains mentioned above and their growth under stress conditions. Moreover, we also tested the role of Vlp1 in the pathogenesis of *Cryptococcus* through a mouse inhalation model of systematic infection and a cryptococcal–macrophage interaction experiment. Our results showed that the Vlp1 is critical for fungal pathogenicity in *C**. neoformans*, as the *vlp1*Δ mutant is avirulent and is gradually eliminated by the host after infection. Thus, the vacuolar morphogenesis protein Vam6-like protein Vlp1 plays an essential role in pathogenicity in *C. neoformans*.

## 2. Materials and Methods

### 2.1. Strains and Culture Conditions

The *C. neoformans* strains used in this study are listed in Table 1. *C. neoformans* strains were routinely cultured on yeast extract peptone dextrose (YPD) medium at 30 °C. The melanin media was used for melanin production of cryptococcal strains [15]. A diluted Sabouraud medium was used for capsule formation induction of cryptococcal strains [16]. YPD medium supplemented with different chemicals was used to test the growth of cryptococcal strains under different stress conditions and prepared as described previously [11].

### 2.2. Generation of vlp1Δ Mutants and Their Complemented and Overexpressed Strains

A split marker strategy was used to knock out the *VLP1* genes in both H99 and KN99**a** strains background [20]. The fusion fragments for *VLP1* gene knockout were obtained through two rounds of PCR. Briefly, in the first round of PCR, the 5′ and 3′ flanking sequences of the *VLP1* gene were amplified with primers TL327/328 and TL329/330 (see Table 2 for primer information), respectively, using the H99 genomic DNA as templates. The dominant selection marker *NEO* was amplified from the plasmid pJAF1 using primers TL17 and TL18 [20,21]. In the second round of PCR, the 5′ or 3′ flanking sequence was mixed with the *NEO* marker as templates. Primers TL327/29 and TL19/330 were used to amplify the 5′ or 3′ fusion fragments of the flanking sequences and *NEO* marker, respectively. Then, the same amount of the 5′ and the 3′ fusion fragments was combined and precipitated onto the 0.6 µm gold microcarrier beads (Bio-Rad, Hercules, CA, USA) and transformed into the wild-type strains H99 and KN99a biolistically. Stable transformants were further selected on YPD plates containing 200 mg/liter G418. The *vlp1*Δ mutants were confirmed by diagnostic PCR using positive primers F4/R4 (TL333/59) and negative primers F3/R3 (TL331/332) and Southern blot analysis.

To generate the *vlp1*Δ mutant complemented strains, a genomic DNA fragment containing the 1.5-Kb upstream promoter, the *VLP1* open reading frame, and the 0.5-Kb downstream terminator was amplified using primers TL1016/1017 and cloned into the pTBL1 plasmid to generate the gene complementary vector pTBL212. The resulting plasmid pTBL212 linearized by *Sca*I was biolistically transformed into the α and **a** mating type *vlp1*Δ mutants. Phenotypic analysis under stress conditions was used to verify whether the phenotypes of the *vlp1*Δ mutant have recovered.

To overexpress the *VLP1* gene, we amplified the ORF of *VLP1* with primers TL1018/1019 using H99 genomic DNA as templates and cloned into the pTBL153 (a vector containing action promoter and HA tag, not published) to generate the *VLP1* overexpression vector pTBL208. The pTBL208 was then linearized with *Sca*I and biolistically transformed into the α and a mating type *vlp1*Δ mutants after sequencing analysis to confirm that the *VLP1* gene was not mutated. Stable transformants were further selected on YPD plates containing 100 mg/liter nourseothricin sulfate. The overexpression of the *VLP1* gene was evaluated and confirmed by quantitative real-time PCR as described previously [11,20]. Briefly, the overnight cultures of each cryptococcal strain were collected, and the pellets were used for total RNA extraction after being washed twice with ddH_2_O. Total RNAs were extracted and purified using an Omega Total RNA kit II (Omega Bio-tek, USA), and the first-strand cDNAs were synthesized using a Hifair^®^ II 1st Strand cDNA Synthesis Kit (Yeasen, Shanghai, China) following the manufacturer’s instructions. The expression levels of the *VLP1* (primers: TL1281/1282, see Table 1 for primers information) gene was normalized using the *GAPDH* (primers: TL217/218) gene as an internal control, and the relative levels were determined using the comparative threshold cycle (C_T_) method [22].

### 2.3. Melanin Production and Capsule Formation Assay

To examine the melanin production of the cryptococcal strains, we grew the cells of each cryptococcal strain in melanin media (MM) and isolated the old and young cells of *Cryptococcus* following the protocol described by Orner et al. [15]. The pigmentation of each cryptococcal strain was measured by histogram analysis using FUJI opensource software as described previously [15,23]. To examine the capsule formation of cryptococcal strains, overnight cultures in YPD of each strain were washed twice with PBS buffer and incubated in the diluted Sabouraud medium overnight at 37 °C [16]. The size of the capsules was measured and analyzed as described previously [11]. The average and standard deviation from at least 100 cells were calculated for each cryptococcal strain tested.

### 2.4. Virulence Studies

To examine the role of Vlp1 in fungal virulence, the overnight cultures of each cryptococcal strain were washed twice with PBS and diluted to a final concentration of 2 × 10^6^ cells per milliliter. Ten female C57 BL/6 mice of each group were intranasally inoculated with 10^5^ cells of each strain as described previously [24]. The mice that appeared moribund or in pain during the animal study were sacrificed by carbon dioxide inhalation. Statistical analysis of the survival data between paired groups of the virulence study was performed using PRISM v.8.0 (GraphPad Software, San Diego, CA, USA).

### 2.5. Fungal Burdens and Histopathology in Infected Organs

In accordance with the Southwest University-approved animal protocol, the *Cryptococcus*-infected mice were sacrificed at the endpoint of the animal experiment, and the animal experiment was terminated at 80 dpi as described previously [20]. To compare the fungal burdens of the mice infected by each cryptococcal strain, we isolated the brains, lungs, and spleens of the infected mice at the endpoint of the experiment and then homogenized them with a homogenizer. One hundred microliters of diluted resuspensions were spread on the YPD plates containing ampicillin and chloramphenicol, and the number of the colonies was determined after incubation at 30 °C for two days. Meanwhile, to compare the degree of destruction of *Cryptococcus* to host organs and the host’s immune response, we fixed the brains, lungs, and spleens with 10% formalin solution and sent them to the Servicebio Biological Laboratory for section preparation (Servicebio, Wuhan, China).

### 2.6. Serum Treatment and Cryptococcus–Macrophage Interaction Assay

To test *Cryptococcus* serum survival, we performed the serum treatment on *Cryptococcus* cells and measured the viability of *Cryptococcus* cells as described previously, with minor modification [25]. Briefly, overnight cultures of each *Cryptococcus* strain were washed twice with sterile distilled water and then added to 450 µL of fetal bovine serum (FBS; Gibco, catalog No. 10099141C) to a final concentration of 1 × 10^6^ cells/mL. The mixture of cryptococcal cells and serum were incubated at 37 °C. At the indicated time points, aliquots were taken out, serially diluted, and spread onto YPD medium to determine cell viability.

The *Cryptococcus*–macrophage interaction test was also performed as previously described [11,12]. J774 macrophages were used in this study, and phagocytosis was allowed to occur for 2 h at 37 °C in 10% CO_2_ before the fresh DMEM was added to wash away the nonadherent extracellular *Cryptococcus* cells. After an additional 0, 2, or 22 h of incubation, distilled water was added to each well to lyse macrophage cells after removing the DMEM. The lysate was spread on YPD plates after appropriate dilution, and yeast CFUs were counted to determine the phagocytosis rate and intracellular proliferation.

### 2.7. Blood Analysis

To test the immune stimulation effects of *vlp1*Δ mutants on mice after *vlp1*Δ mutant infection, we collected blood from mice used to analyze the progression of *vlp1*Δ mutant infection. Serum cytokines of 1 d, 7 d, and 14 d infection experiments (IL-2, IL17A, IL-19, and IFN-γ) and the non-infected mice were measured with ELISA using the mouse ELISA kit (LMAI Bio, Shanghai, China). The serum cytokine determination procedure is as follows: after setting up the standard and testing sample wells, 50 µL of standard and 40 µL of sample dilution were added to the standard and testing sample wells, respectively. Then, 10 µL testing samples were added to each testing sample well and mixed gently. After that, 100 µL of HRP-conjugate reagent was added to each well, and the plate was incubated at 37 °C for 60 min after closing with the closure plate membrane. Next, the liquid in each well was discarded, and each well was washed five times with washing buffer and dried by pat. Finally, the reaction was visualized by adding 50 µL of chromogen solution A and B and the stop solution. The plates were read at 450 nm wavelength with a microplate reader.

### 2.8. Ethics Statement

The animal experiments conducted at the Southwest University were in complete accordance with the “Guidelines for the Ethical Care of Laboratory Animals (No. 398, 2006)” of the Ministry of Science and Technology of China. All vertebrate experiments were approved by the Animal Ethics Committee of Southwest University (IACUC-20190306-07, 06/03/2019).

## 3. Results

### 3.1. Identification of Vlp1 in C. neoformans

In our previous study, an iTRAQ-based proteomics approach (Applied Protein Technology, Shanghai, China) was used to identify the downstream substrates of the F-box protein Fbp1, a key component of the E3 ligase complex, in *C. neoformans* [14]. One of the candidates, CNAG_05395, was expressed in high abundance in the background of *fbp1*Δ mutants and contained three PEST domains in its sequence, which may be a downstream substrate of Fbp1. Since Fbp1 has a critical function in *Cryptococcus*, its function may be achieved by regulating its downstream substrates; it is very necessary to conduct a functional analysis of CNAG_05395 in *C. neoformans*. After querying the FungiDB database [26], we found that the CNAG_05395 gene is 3301 bp in length, contains 12 exons, and encodes a 1044 amino acid protein. Protein sequence analysis showed that the CNAG_05395 protein has one clathrin heavy chain (CNH) domain, one vacuolar sorting protein 39 domain 1, and one vacuolar sorting protein 39 domain 2 [27,28,29] (Figure 1A). Sequence blast analysis showed that CNAG_05395 protein has a 38% sequence similarity to the vacuolar morphogenesis protein Vam6 in *Saccharomyces cerevisiae* [28] (Figure 1B,C); we, therefore, name it vacuolar morphogenesis protein Vam6-like protein Vlp1 in *C. neoformans*. Since Vam6 plays a vital role in vacuolar morphogenesis in *S. cerevisiae* and may also play an important role in *C. neoformans*, we decided to investigate the function of Vlp1 in *C. neoformans*.

### 3.2. Disruption, Complementation and Overexpression of VLP1

To investigate the role of Vlp1 in *C. neoformans*, we generated the *vlp1*Δ mutants in both α and a mating type of the wild-type strains backgrounds using a split marker strategy. Stable transformants were first screened by diagnostic PCR using positive and negative primers (Appendix A). Then, the same amount of genomic DNAs of the transformants verified by PCR were digested with the restriction enzyme *Sac*I, fractionated, and hybridized with a *VLP1* downstream flanking sequence-specific probe (Appendix A). The wild-type strain H99 generates a 7.0-Kb band, while the *vlp1*Δ mutants generate a 3.0-Kb band (Appendix A). The *Sac*I linearized complementation vector pTBL212 was transformed into the *vlp1*Δ mutants, and the growth of the transformants under stress conditions was detected to determine whether the gene complementation was successful (data not shown). The overexpression of *VLP1* was verified by RT-qPCR (Appendix A).

### 3.3. Disruption of VLP1 Affects Melanin Production and Capsule Formation in C. neoformans

*C. neoformans* has three classic virulence factors: melanin production, capsule formation, and growth at 37 degrees, which helps *Cryptococcus* to infect the host. To evaluate the role of Vlp1 in the formation of cryptococcal virulence factors, we tested the melaninization, capsulation, and growth of the *vlp1*Δ mutants at 37 °C in vitro. To quantify melaninization, cell pellets of each *Cryptococcus* strain were imaged and assessed using histogram analysis to determine the intensity value on the black to white scale. Compared with the wild-type strain H99, the melanin produced by the young cells of the *vlp1*Δ mutant (IV = 190) and *VLP1* overexpression strain *VLP1*^OE^ (IV = 191) was less than that of the wild type (IV = 166) (Figure 2A). Interestingly, old *vlp1*Δ (IV = 185) and *VLP1*^OE^ (IV = 128) cells melanized much less than that of the wild-type H99 (IV = 73). H99 young (IV = 166) and old (IV = 73) cells showed a shift of 93 in magnitude while the *vlp1*Δ young (IV = 190) and old (IV = 185) cells and the *VLP1*^OE^ young (IV = 191) and old (IV = 128) cells showed a shift of 5 and 63 in magnitude, respectively.

To evaluate the role of Vlp1 in the formation of cryptococcal capsules, we used the diluted Sabouraud medium to induce the formation of capsules. Our results showed that the deletion of the *VLP1* gene resulted in a reduced capsule size in *Cryptococcus* (Figure 2B). However, interestingly, under the same culture conditions, overexpression of the *VLP1* gene will cause the enlargement of the capsule in *Cryptococcus* (Figure 2B). Statistical analysis showed that the difference between the size of the capsule formed by the *vlp1*Δ mutant or the *VLP1* overexpression strains and that of the capsule formed by the wild-type strain was extremely significant (Figure 2C, *p* < 0.001 and *p* < 0.0001, respectively, Student’s *t*-test), indicating that the Vlp1 proteins played an important role in the formation of the cryptococcal capsules. When the *VLP1* gene was re-transformed into the *vlp1*Δ mutants, the melanin and capsule size of the *vlp1*Δ mutants was restored, indicating that the change in the melanin and capsule production of the *vlp1*Δ mutants was indeed caused by the knockout of the *VLP1* gene.

### 3.4. Vlp1 Is Required for Membrane Integrity

To evaluate the role of Vlp1 in the growth of *C. neoformans* under different stress conditions, we tested the growth of *Cryptococcus* strains under stress conditions such as osmotic stress (1.5 M KCl, 1.5 M NaCl, and 1.5 M sorbitol), nitrosative stress (1 mM NaNO_2_, pH = 4.0), and cell integrity stress (0.025% SDS, 0.5% Congo red, and 250 µg/mL CFW). Our results showed that the *vlp1*Δ mutants are sensitive to SDS and NaCl but not to Congo red or CFW, indicating that Vlp1 may regulate the cell membrane integrity of cryptococcal cells. Meanwhile, the *vlp1*Δ mutant and the *VLP1*^OE^ strain are also sensitive to NaCl, which seems to imply that the Vlp1 protein is necessary for *Cryptococcus* to respond to high salt environments (Figure 2D).

### 3.5. Vlp1 Is Essential for Fungal Infection

To evaluate the role of Vlp1 in the virulence of *C. neoformans*, we tested the virulence of *Cryptococcus* strains with a mouse inhalation model of cryptococcosis. Female C57 BL/6 mice (10 mice per group) were intranasally infected with 10^5^ yeast cells of the wild-type, *vlp1*Δ mutants, *VLP1* complemented, or overexpressed strains. All infected mice were checked twice a day to monitor signs of morbidity. Mouse survival curves showed that the mice infected with wild-type or *VLP1* complemented strains died between 17 and 28 days post infection (dpi) (*p* > 0.15). Interestingly, the mice infected with the *vlp1*Δ mutant survived until the end of the experiment (80 days after infection) and remained healthy (*p* < 0.0001, Figure 3A). Compared with wild-type strains, the virulence of the *VLP1*^OE^ strains was also significantly reduced as the mice infected with *VLP1*^OE^ strains survived between 30 and 35 dpi (*p* < 0.0001, Figure 3A). The above results indicated that Vlp1 plays an essential role in the virulence of *C. neoformans*.

To further explore the reasons for the virulence defects of *vlp1*Δ mutants, we examined the fungal loads in the organs of infected animals when the infection experiment was terminated. Yeast cells were recovered from the brains, lungs, and spleens of five mice infected by each strain, and the fungal loads of these organs were evaluated in terms of yeast colony forming units (CFUs) per gram of fresh organs. Surprisingly, no yeast cells were recovered from the brains, lungs, and spleens of mice infected with the *vlp1*Δ mutants (Figure 3B, *p* < 0.0001), which indicates that the *vlp1*Δ mutants had been eliminated by the host when the experiment was terminated. Moreover, the mice infected by the *VLP1*^OE^ strains had a higher fungal load in the brains than that of the mice infected by the wild-type strains (Figure 3B, *p* < 0.0001). These data further indicated that Vlp1 plays an essential role in fungal pathogenicity in *C. neoformans.*

Next, we prepared H&E-stained sections of the brains, lungs, and spleens of mice infected with *Cryptococcus* strains and examined the development of fungal lesions in the organs. Severe lesions were caused in the brains, lungs, and spleens of the mice infected by the wild-type and the complemented *vlp1*Δ::*VLP1* strains at 23 dpi (Figure 3C). The *VLP1*^OE^ strains also caused severe lesions in the organs of the mice at the end of the experiment (80 dpi). In contrast, the *vlp1*Δ mutants could not generate any visible lesions, even when the experiment was terminated (Figure 3C). Taken together, our results showed that Vlp1 is essential for the fungal pathogenicity of *C. neoformans*.

### 3.6. Vlp1 Is Essential for Progression of Fungal Infection

Our above results indicated that the host would eliminate the *vlp1*Δ mutants after the inoculation of mice. To better understand the dynamics of the *vlp1*Δ mutant–host interaction during the progression of infection, we examined the fungal loads in the organs and observed the development of fungal lesions on H&E-stained sections at 1, 3, 5, 7, 14, and 21 dpi.

At 1–5 dpi, *Cryptococcus* yeast cells were not recovered from the brains of the mice infected by the wild-type or the *vlp1*Δ::*VLP1* strains (Figure 4A). From 7 dpi, yeast cells were gradually recovered from the brains of wild-type or the *vlp1*Δ::*VLP1* infected mice (Figure 5A). In contrast, yeast cells could not be recovered from the brains of the mice infected by the *vlp1*Δ mutants at 1–21 dpi, indicating that the *vlp1*Δ mutants cannot infect the brains after intranasal infection (Figure 4A). Meanwhile, the mice infected with the wild-type or the *vlp1*Δ::*VLP1* strains suffered cryptococcosis, and their brains were severely damaged, while the *vlp1*Δ mutant-infected mice were healthy, and no lesions were found in their brains (Figure 4D, left panel). At 1 to 21 dpi, the fungal loads recovered from the lungs of mice infected with wild-type or the *vlp1*Δ::*VLP1* strains gradually increased, while that of the *vlp1*Δ mutants showed a gradual decrease, and no yeast cells were recovered from the *vlp1*Δ mutant-infected lungs after 7 dpi (Figure 4B). The results of H&E-stained slides observation showed that the lung damage caused by the wild-type or the *vlp1*Δ::*VLP1* strains infection was aggravated with the continuous proliferation of cryptococcal cells, while the lungs infected by the *vlp1*Δ mutants were relatively intact (Figure 4D, middle panel). The fungal loads recovered in the spleens of each cryptococcal strain-infected mouse is similar to that of the brains; that is, the spleen infected with the wild-type or the *vlp1*Δ::*VLP1* strains can be recovered from the yeast cells and gradually increase from 7 days; while the *vlp1*Δ mutants-infected spleen did not recover yeast cells at 1–21 dpi (Figure 4C,D).

### 3.7. Vlp1 Is Important for Proliferation inside Macrophage and Survival in the Host Complement System

The results of the virulence study above showed that the *vlp1*Δ mutants were gradually eliminated by mice in a murine inhalation model of cryptococcosis. To figure out the reason for the elimination of *vlp1*Δ mutants, we first performed *Cryptococcus*–macrophage interaction assays and tested the survival and proliferation of *vlp1*Δ mutants in the J774 murine macrophage cells, since macrophages are the first type of cells encountered by *C. neoformans* after entering the host lungs. After two or four hours of coincubation, the number of yeast CFUs recovered from the macrophages co-incubated with *vlp1*Δ mutants was comparable to that recovered from the wild-type or the complemented strain-interacting macrophages, suggesting a similar phagocytosis level between the wild-type and the *vlp1*Δ mutants (Figure 5A). However, after 24 h of incubation, significantly fewer yeast CFUs were recovered from the macrophages infected by *vlp1*Δ mutants (*p* < 0.0001) (Figure 5A). These data indicated that once the *vlp1*Δ mutant is phagocytosed by macrophages, its proliferation rate is slower than that of the wild type, which may be one reason for the loss of the virulence of the *vlp1*Δ mutant in the murine systemic infection model.

Meanwhile, to verify whether the components of the host complement system can cause damage to *C. neoformans*, we also examined the viability of cryptococcal cells incubated with mouse serum for 1, 2, 3, and 4 h. Our results showed that, after 4 h of incubation of cryptococcal cells with the mouse serum, the survival rate of the *vlp1*Δ mutant was significantly lower than that of the wild type (*p* < 0.001, Figure 5B).

The above results suggested that the Vlp1 protein plays a vital role in the intracellular proliferation of *Cryptococcus* in macrophages or the viability of *Cryptococcus* in the host complement system.

### 3.8. Vlp1 Is Not Required to Stimulate the Immune Response in Blood

To detect whether *vlp1*Δ mutant infection can cause changes in the host’s immune response in blood, we collected the blood from mice while sampling the organs for the progression of fungal infection. The serum cytokines were measured by a mouse ELISA kit. We compared the cytokine production at 1, 7, and 14 days after infection. We observed that the deletion of the cryptococcal *VLP1* gene did not affect the production of cytokines such as IL-2, INF-γ, Il-17A, and IL-18 in host blood in the early, middle, and late stages of infection (Appendix A). Thus, in this experiment, the expression of cryptococcal *VLP1* may not affect the immune responses in the blood of the host.

## 4. Discussion

Cryptococcal infection has become one of the common complications in immunodeficient or immunocompromised patients and is also a significant cause of death. However, due to the lack of therapeutic drugs for *C. neoformans* and the objective fact that fungi are resistant to the drugs [30], the research on the pathogenesis and the mechanism of drug resistance of *C. neoformans* has become a hot and difficult point in current research.

As the key component of E3 ubiquitin ligase, F-box protein is responsible for the recognition and ubiquitination of downstream targets and plays a crucial role in the morphological development and pathogenicity of fungi. In *S**. cerevisiae*, the F-box proteins Cdc4 and Grr1 regulate multiple processes in the cell by regulating the degradation of various proteins [9]. Our previous study identified a cryptococcal F-box protein Fbp1 and showed that Fbp1 is essential for the sexual reproduction and virulence of *C. neoformans* [11,12,13]. However, it is still unknown how many substrates cryptococcal Fbp1 has and how Fbp1 regulates the morphological development and virulence of *C. neoformans* through these targets. In the previous study, we tried to identify the downstream targets of Fbp1 using an iTRAQ-based proteomics approach and found that Vlp1 could be a potential downstream target of Fbp1 [14]. In the present study, we identified the *Cryptococcus* Vlp1 and revealed that Vlp1 is essential for the virulence of *C. neoformans*.

*Cryptococcus* Vlp1 shares 38% similarity with the Vam6 protein in *S*. *cerevisiae* (Figure 1B). In *S*. *cerevisiae*, Vam6 is a component of a protein complex that resides on the vacuolar membranes and involves the last step of the vacuolar assembly [28]. In addition to defects in vacuole morphology development, null mutants of Vam6 are sensitive to high temperature (37 °C) [31]. However, *Cryptococcus* Vlp1 does not regulate heat sensitivity in *C. neoformans*, as *vlp1*Δ mutant grows normally at 37 °C like the wild-type strain (Figure 2C). It is possible that, as a human fungal pathogen, *C. neoformans* has adapted to the human body temperature in the evolutionary process, and the deletion of the *VLP1* gene will not affect the tolerance of *C. neoformans* to high temperature. Another possibility is that although *Cryptococcus* Vlp1 has a 38% similarity to the Vam6 in *S*. *cerevisiae*, the similarity is not high enough to explain the functional similarity of the two, and the loss of the pathogenicity of *Cryptococcus vlp1*Δ mutants also illustrates this point.

*C. neoformans* has three classical virulence factors: melanin, polysaccharide capsule, and growth at body temperature, which favor the infection and survival in the host [32,33,34]. In this study, both deletion and overexpression of the *VLP1* gene resulted in a decrease in the ability of cryptococcal cells to melanize (Figure 2A). Given the possible differences in melanization between the new and old cells of *Cryptococcus*, we have tested the melanin production of the new and old cells in each *Cryptococcus* strain. The results showed that the ability of melanin production of the *vlp1*Δ mutants and the *VLP1*^OE^ strains was indeed lower than that of the wild-type strain, regardless of the new cells or old cells, which may also be one of the reasons for the reduced pathogenicity of *Cryptococcus* caused by *VLP1* deletion or overexpression.

Our results also showed that the *VLP1* deletion resulted in a smaller *Cryptococcus* capsule, while the *VLP1* overexpression resulted in a larger capsule of *Cryptococcus* (Figure 2B,C), suggesting that *VLP1* may be involved in the formation of the *Cryptococcus* capsule. In *C. neoformans*, there are several signaling pathways involved in regulating the formation of the capsule. The disruption of the *PKA1* gene in the Gpa1–cyclic AMP (cAMP) pathway resulted in the failure of melanin or capsule production and the loss of the pathogenicity of *C. neoformans* [35]. Deleting the *PKC1* of the protein kinase C (PKC) pathway also led to alterations in the capsule and melanin of *C. neoformans* [36]. Additionally, disruption of the *HOG1* gene in the Hog1 mitogen-activated protein (MAP) kinase pathway will affect the melanin and capsule production in serotype A strains of *C. neoformans* [37]. All the above three signaling pathways are related to regulating the production of capsule and melanin of *C. neoformans*. Our results also indicated that both the *VLP1* disruption and overexpression affect the production of capsule and melanin of *C. neoformans*. It is interesting to observe that Vlp1 is involved in capsule formation and melanin production in *C. neoformans*. Whether the function of Vlp1 is related to any of the signaling pathways mentioned above remains unknown and needs to be further explored.

Our study showed that the deletion of the *VLP1* gene resulted in the complete loss of the pathogenicity of *C. neoformans* (Figure 3). To understand the cause for the loss of pathogenicity of the *vlp1*Δ mutant, we carried out the fungal infection progression assay and the *Cryptococcus*–macrophage interaction assay. Progression of *vlp1*Δ mutant infection in vivo at 1–21 dpi showed that the *vlp1*Δ mutant was gradually eliminated by the host lung at 7 dpi and could not disseminate to the brains and spleens (Figure 4), which explains why the *vlp1*Δ mutant loses its pathogenicity in a murine systematic model of cryptococcosis. Macrophages are the first type of cells encountered by *C. neoformans* after entering the host lungs. To figure out how the *vlp1*Δ mutant is cleared by the host lungs, we performed the *Cryptococcus*–macrophage interaction assay and *Cryptococcus* survival assay in the host complement system. The results of the *Cryptococcus*–macrophage interaction assay showed that the *vlp1*Δ mutant had intracellular proliferation defects inside macrophages (Figure 5A). Besides, the viability assay of *Cryptococcus* in mouse serum revealed that the survival rate of the *vlp1*Δ mutant was significantly lower than that of the wild type (Figure 5B). The results of the above two experiments suggested that the loss of Vlp1 significantly affects the proliferation or survival of *Cryptococcus* in host cells, which could be one reason why the *vlp1*Δ mutant is eliminated in the mouse systemic infection model.

Our previous studies showed that although the F-box protein Fbp1 is not involved in the development of classical virulence factors, the deletion of the *FBP1* gene leads to a decrease in cell membrane integrity of *Cryptococcus* cells and defects in intracellular proliferation after phagocytosis by macrophages, resulting in the complete loss of the pathogenicity of *fbp1*Δ mutants. This study showed that the deletion of the *VLP1* gene resulted in almost the same phenotypes as the *fbp1*Δ mutants: low cell membrane integrity, defects in macrophage proliferation, and loss of pathogenicity. Given that Vlp1 is a high-abundance protein in the background of *fbp1*Δ mutants and the phenotypes of *vlp1*Δ mutants and *fbp1*Δ mutants are consistent, we speculate that Vlp1 may be a downstream target of Fbp1. However, the current evidence is insufficient to prove that Vlp1 is a downstream target of Fbp1.

It would be very interesting to prove that Fbp1 regulates the virulence of *C. neoformans* through the downstream target Vlp1. Further studies such as the interaction between Fbp1 and Vlp1, the stability and/or ubiquitination of Vlp1 under the control of Fbp1 are still needed. Overall, our study demonstrated the vacuolar morphogenesis protein Vam6-like protein Vlp1 that is essential for the pathogenicity of *C. neoformans*.

## Figures and Tables

**Figure 1 jof-07-00418-f001:**
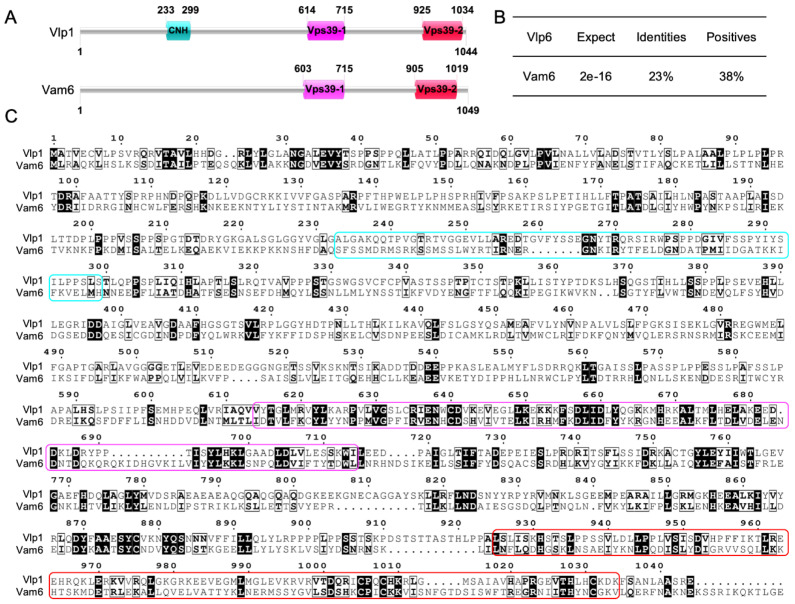
Sequence analysis of Vlp1 in *C. neoformans*. (**A**) Cryptococcal Vlp1 contains one clathrin heavy chain (CNH) domain, one vacuolar sorting protein 39 domain 1 and one vacuolar sorting protein 39 domain 2; (**B**) Cryptococcal Vlp1 shows a 23% identity and 38% similarity with the Vam6 protein in *S. cerevisiae*; (**C**) Sequence alignment of Vlp1 in *C. neoformans* and Vam6 in *S. cerevisiae*. The cyan round rectangular box shows the sequence of the CNH domain; the purple and red round rectangular box shows the sequences of the Vps39-1 and Vps39-1 domains, respectively.

**Figure 2 jof-07-00418-f002:**
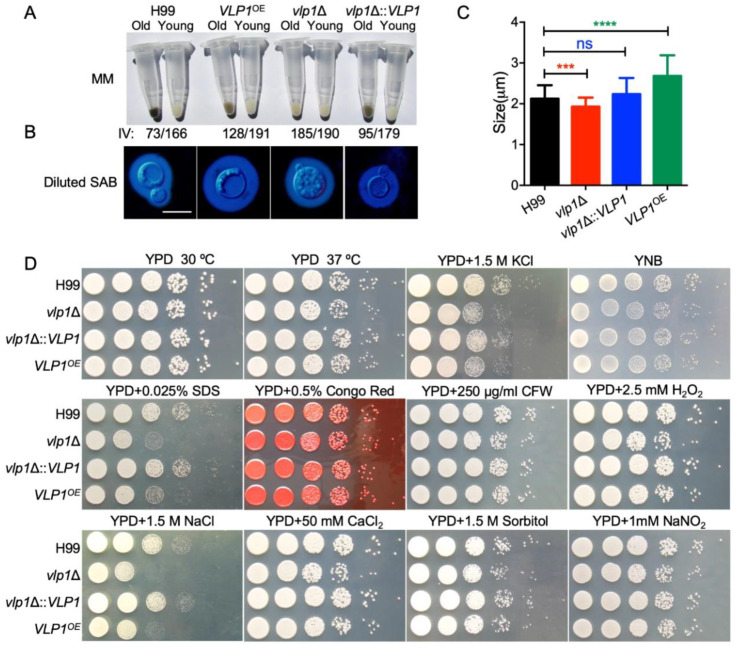
Disruption of *VLP1* affects the capsule formation, melanin production, and growth under stress conditions in *C. neoformans*. (**A**) Melanin production of each *Cryptococcus* strain was induced in melanization media. Cell pellets (top) of the old or young cells were imaged, and the darkness (bottom) of the cells was quantified using black to white histogram analysis. (**B**) Capsule formation of each *Cryptococcus* strain was examined using India ink staining under a microscope after overnight culture (~18 h) in diluted Sabouraud medium. SAB: Sabouraud medium; bar, 10 µm. (**C**) Statistical analysis of the capsule formation in diluted Sabouraud medium. At least 100 cells were measured for each *Cryptococcus* strain, and the data are shown as the mean ±SD from three repeats. ns: not significant; ***, *p* < 0.001; ****, *p* < 0.0001. (**D**) Growth of each *Cryptococcus* strain under different stress conditions. Overnight culture of each *Cryptococcus* strain was first adjusted to OD600 value of 2.0 and then 10-fold serially diluted with ddH_2_O. An amount of 5 µL of each dilution was dropped onto the YPD plates with different stresses and then incubated at 30 °C for 2 or 3 days. The names of the *Cryptococcus* strains are labeled on the left, and the culture conditions are shown on the top.

**Figure 3 jof-07-00418-f003:**
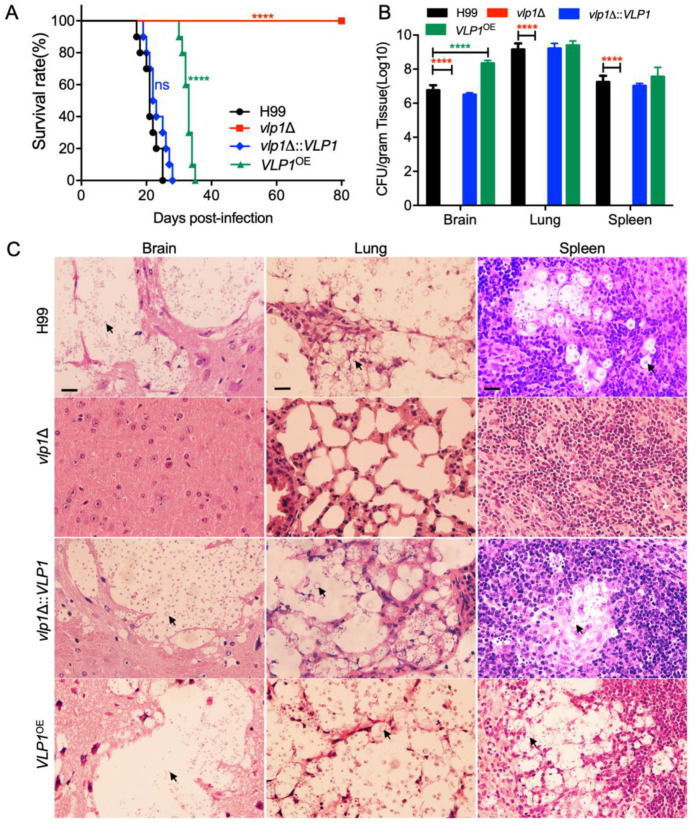
Vlp1 regulates fungal pathogenicity in *C. neoformans*. C57 BL/6 mice (10 mice per group) were used to examine the pathogenicity of each cryptococcal strain. (**A**) Survival curve of C57 BL/6 mice infected with the wild-type, *vlp1*Δ mutant, *VLP1* complemented, and overexpressed strains. The *vlp1*Δ mutant is completely avirulent compared with the wild type. ns: not significant; ****, *p* < 0.0001 (determined by the log-rank (Mantel–Cox) test). (**B**) Fungal loads in brains, lungs, and spleens of *Cryptococcus*-infected mice at the end of the experiment. The data shown are mean ± SD of five mice. ****, *p* < 0.0001 (determined by Mann–Whitney test). (**C**) H&E-stained slides from the cross-sections of the organs were prepared and observed under a light microscope when the experiment was terminated. The cryptococcal cells are indicated by the arrow. Bars, 20 µm.

**Figure 4 jof-07-00418-f004:**
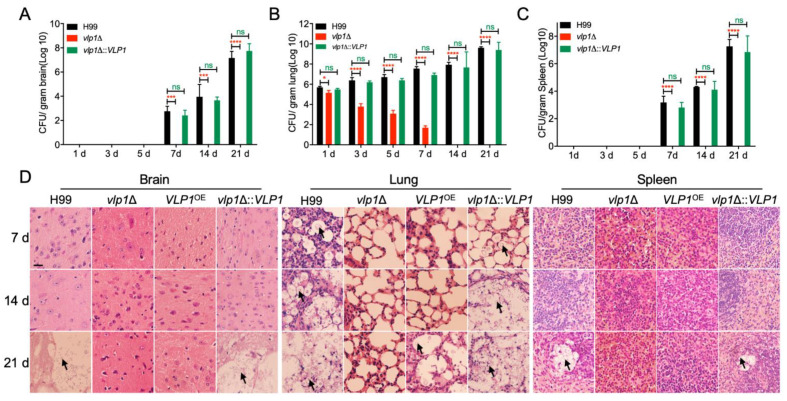
Progression of fungal infection in *vlp1*Δ mutant infected organs. (**A**) Organs of five mice infected by each cryptococcal strain were isolated at 7, 14, and 21 dpi. CFU of each fresh organ was measured in the brain (**A**), lung (**B**), and spleen (**C**) homogenates. The data point shown is the mean ± SD for values from five mice. ns: not significant; *, *p* < 0.05; ***, *p* < 0;001; ****, *p* < 0.0001 (determined by Mann–Whitney test). (**D**) H&E-stained slides from the cross-sections of the organs prepared and observed under a light microscope. The cryptococcal cells are indicated by the arrow. Bars, 20 µm.

**Figure 5 jof-07-00418-f005:**
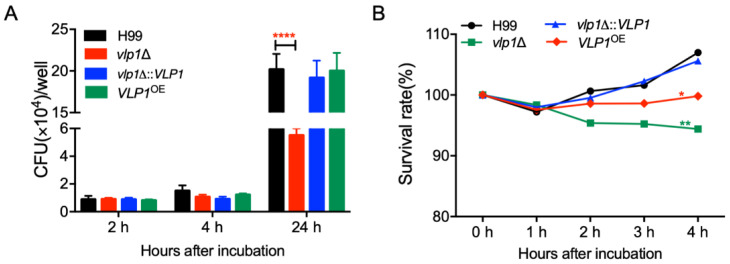
Vlp1 is required for *Cryptococcus*–macrophage interaction and survival in the host complement system. (**A**) The proliferation of *Cryptococcus* inside macrophages was performed using J774 macrophages. CFU numbers recovered from the macrophage culture after removing the nonadherent extracellular yeast cells and cultured for indicated times were used to determine intracellular proliferation and macrophage killing. ****, *p* < 0.0001 (determined by Mann–Whitney test). (**B**) The survival rate of the same strains as (**A**) after co-incubation with mouse serum was determined by the CFU numbers recovered from the co-incubation cultures after co-incubated as the indicated time. *, *p* < 0.05; **, *p* < 0.001.

**Table 1 jof-07-00418-t001:** Strains and plasmids used in this study.

Strains and Plasmids	Genotype or Properties	Source or Reference
*C. neoformans*		
H99	*MAT*α	Perfect et al., 1993 [17]
KN99**a**	*MAT* **a**	Nielsen et al., 2003 [18]
TBL131	*MAT*α *vlp1*Δ::*NEO*	In this study
TBL268	*MAT***a** *vlp1*Δ::*NEO*	In this study
TBL278	*MAT*α *vlp1*Δ::*NEO GFP-VLP1*::*NAT*	In this study
TBL342	*MAT***a** *vlp1*Δ::*NEO GFP-VLP1*::*NAT*	In this study
TBL350	*MAT*α *vlp1*Δ::*NEO VLP1-HA*::*NAT*	In this study
TBL351	*MAT***a***vlp1*Δ::*NEO VLP1-HA*::*NAT*	In this study
TBL381	*MAT*α *vlp1*Δ::*NEO VLP1-HA*::*NAT*	In this study
TBL382	*MAT***a***vlp1*Δ::*NEO VLP1*::*NAT*	In this study
Plasmids		
pCN19	Amp^r^ Plasmid harboring *GFP* under histone H3 promoter	Price et al., 2008 [19]
pTBL1	Amp^r^ Plasmid harboring *NAT* marker	Fan et al., 2019 [20]
pTBL20	Amp^r^ Vector for *P_ACTIN_-ZFP1HA-NAT* for *ZFP1* overexpression	Fan et al., 2019 [20]
pTBL155	Amp^r^ Vector for *P_H3_-GFP-VLP1* for *VLP1* localization	In this study
pTBL187	Amp^r^ Vector for *P_H3_-GFP-VLP1* for *VLP1* localization	In this study
pTBL208	Amp^r^ Vector for *P_ACTIN_-VLP1-NAT* for *VLP1* overexpression	In this study
pTBL212	Amp^r^ Vector for *P_VLP1_-VLP1-NAT* for *VLP1* complementation	In this study

**Table 2 jof-07-00418-t002:** PCR primers used in this study.

Primers	Targeted Genes	Sequence (5′-3′)
TL17	M13F	GTAAAACGACGGCCAG
TL18	M13R	CAGGAAACAGCTATGAC
TL19	*NEO* split F	GGGCGCCCGGTTCTTTTTGTCA
TL20	*NEO* split R	TTGGTGGTCGAATGGGCAGGTAGC
TL59	*NEO* R4	TGTGGATGCTGGCGGAGGATA
TL217	*GAPDH* Q-PCR F1	TGAGAAGGACCCTGCCAACA
TL218	*GAPDH* Q-PCR R1	ACTCCGGCTTGTAGGCATCAA
TL327	*VLP1* KO F1	GGTCAGGCGTGGAAGCGTCATACA
TL328	*VLP1* KO R1	CTGGCCGTCGTTTTACGCTGCAACAAGTCGCGTCATTTAC
TL329	*VLP1* KO F2	GTCATAGCTGTTTCCTGTCCGCAATGCCACAAGAGACTG
TL330	*VLP1* KO R2	GAGACCCGGGGCCTAATACCTAAT
TL331	*VLP1* KO F3	GCTCACCCCGAAAACAGATACAGG
TL332	*VLP1* KO R3	AAGGCGGGGAGGGAGGATTC
TL333	*VLP1* KO F4	AGCTGTGCCCCATCTTTTTAGTTA
TL971	*VLP1* CDS F1	GACGAGCTGTAcGGATCCATGGCGACCGTCGAGTGCGTGCTG (*Bam*HI)
TL972	*VLP1* CDS R1	CTGGCGGCCGTTACTAGTCTATTCTCTTGAAGCGGCCAAGTT(*Spe*I)
TL1016	*VLP1* Comp F1	GATATCGAATTCCTGCAGCCCGGGGGATCCGGAAAAGTTAAAAGTCATTGGCAGTT (*Bam*HI)
TL1017	*VLP1* Comp R1	CGGTGGCGGCCGCTCTAGAACTAGTGGATCTTGGAAAAGAAAAGGAAAGTGAAATC (*Bam*HI)
TL1018	*VLP1-HA* F1	CGCCCAACATGTCTGGATCCATGGCGACCGTCGAGTGCGTGCTG (*Bam*HI)
TL1019	*VLP1-HA* R1	ACGTCGTATGGGTAGGATCCTTCTCTTGAAGCGGCCAAGTTTGC (*Bam*HI)
TL1281	*VLP1* Q-PCR F1	GAAAGGGAGGAAAGAGGAAGTG
TL1282	*VLP1* Q-PCR R1	TATTCTCTTGAAGCGGCCAAG

## Data Availability

All relevant data are within the manuscript.

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
