# Peer review of "The Vacuolar Morphogenesis Protein Vam6-Like Protein Vlp1 Is Required for Pathogenicity of Cryptococcus neoformans"

_jof, 2021, doi:10.3390/jof7060418_

Round 1
Reviewer 1 Report
5/9/21
The Vacuolar Morphogenesis Protein Vam6 Regulates Virulence of Cryptococcus neoformans.
The authors have identified a putative vacuolar protein, Vam6, from a screen targeting potential E3 ligase substrates. After performing some initial phenotypic characterization and stress tolerance assays, the authors show that Vam6 is essential for virulence. Though the experiments and data appear sound, mechanistic explanations are lacking and some of the data interpretation is tenuous. The only major phenotype identified was sensitivity to high salt concentration. The authors fail to provide direct evidence that C. neoformans Vam6 localizes to the vacuole or plays a role in vacuolar morphogenesis. Instead, these roles are assumed based on homology to Saccharomyces cerevisiae Vam6.
The authors claim that C. neoformans Vam6 is a vacuolar morphogenesis protein, but do not provide any protein localization or vacuolar morphology data to support that claim.
The authors’ claims that Vam6 is required for capsule formation and is dispensable for melanization are somewhat dubious:
Figure 3B: The difference in capsule size, though statistically significant, is incredibly slight. It’s a bit of a stretch to claim that VAM6 is required for capsule synthesis when the difference between wild-type and VAM6 mean capsule thickness looks to be ~0.1 microns.
Figure 3C: Also, even though the authors claim that VAM6 does not influence melanization, it appears that the VAM6 deficient and VAM6 overexpression strain display reduced melanization at 30C and 37C.
Reviewer 2 Report
The article by Fan et al characterizes a novel Cryptococcus protein Vam6 and links it with virulence. Here are my comments on the paper
1)Line 101: What parameters used and how overexpression of VAM6 was quantified with real time PCR? Did the authors use reverse transcriptase (qRT-PCR)?If so mention the parameters that include how RNA was wuantified, cDNA prepared, etc. What oligos used?
2)Did the authors sequenced the overexpressed constructs to verify that no mutations in Vam6 were generation during the cloning process? Please clarify
3)Fig 2: It is better if this figure is put as a
supplemental figure and not as a main figure. The figure is important to know about the generation of the mutant but does not add to the story of the article.
4)Fig 3: It seems that both vam6 del and Vam6 OE melanizes less (Lighter shade) (Fig 3C). Did the authors quantify the melanization? Simple experiment as shown in https://www.frontiersin.org/articles/10.3389/fmicb.2019.02513/full can be used to quantify melanization. It will be better for the paper if the authors can quantify the melanization.
5)SDS and NaCl are osmotic stressers. Did the authors see any
differences in cell wall structures?
6)Fig 3C: L-dopa is a melanin inducing agent. It is confusing as to why the figure is present in place of the stressers. Please clarify
7)Line 216: "DO600" should be OD600.
8)Line 225: "Cryptococcus" needs to be italicized
9)Figs 4 and 5: Red arrows are difficult to visualize. It will be better if the authors change the color of the red arrow.
10)Fig 5: Did the authors perform statistical analysis to determine the p-values in CFUs? If so the significance needs to be shown. Please clarify
11)Fig 5: Since the revertant was already available, why didnt the authors used the revertant as the control for these experiments. Please clarify.
12)Line 301: "Survival of the Host Compliment System"-- all alphabets should be in small letters.
13)I feel figure 7 can go in the supplemental
Reviewer 3 Report
This study describes the characterization of the Vam6 gene in C. neoformans. Previous work by these authors identified the Fbp1 protein that is involved in virulence. The Vam6 gene was identified as a potential substrate of Fbp1. Alteration of Vam6 expression changes capsule size, which results in increased survival in a mouse model after Vam6 deletion. In addition, the vam6∆ strain is killed by host complement, which also contributes to clearance of this mutant.
Comments:
- In general, this is a thorough study characterizing this gene. For the Discussion section, I was wondering if there are any other mutants in the literature that have similar phenotypes that you could compare the Vam6 phenotype to? For example, do mutants in the Gpa1-cAMP, the PKC or the Hog1 MAPK pathways have similar phenotypes as the vam6∆ strain? I suggest adding a discussion of this to the Discussion section.
- Also, did you test whether secretion of GXM was altered in the vam6∆ strain?
- I suggest rewriting the last few sentences of the Introduction, as generally a recapitulation of the results is discouraged in the Introduction. Instead you can briefly list the broad methods that were used to investigate this gene.
- While you state that at least 100 cells were measured to determine capsule size in the figure legend of Figure 3, please also add this detail to the methods section.
- Under section 2.8, please add the protocol number of your approved protocol.
- In the legend for Figure 2, a description of Figure 2C appears to be missing. Please add this to the legend.
- For Figure 4, while you state that the fungal loads were examined at the end of the experiment in the text, this is not stated in the figure legend for (B), please add this detail to the figure legend.
- The heading for section 3.7 states “Cmp1 is Important….”. Is “Cmp1” supposed to be “Vam6”? Please clarify.
- For the macrophage interaction assay, the methods and the figure legend for Figure 6 state that you used J774 macrophages, but the results section states that you used RAW246.7 macrophages? Which did you use? Did you use both? Please clarify.
- While overall the paper is very well written, there are a few grammar mistakes. I suggest having the paper read/edited by a native English speaker.
Minor comments:
- On line 97, “Action” should be “Actin”.
- On line 99, please italicize “vam6∆”.
- On line 216, “DO600” should be “OD600”.
Round 2
Reviewer 1 Report
I cannot recommend that the article be accepted in its present form. I suggest that the authors characterize Vam6 in further detail and provide mechanistic data to support their current claims.
The following are major claims that are still not supported by evidence following the authors' revisions:
1) The authors claim that VAM6 is a vacuolar protein, and in their response to the initial review write: "We claimed that the C. neoformans Vam6 (CNAG_05395) is a vacuolar morphogenesis protein based on the sequence
alignment analysis. Sequence alignment warrants investigation into whether C. neoformans Vam6 is indeed a vacuolar morphogenesis protein, but is not sufficient to claim that it is. This is a major flaw in the paper and makes it difficult to place the following characterization of Vam6 mutants into proper context.
2) It's commendable that the authors improved their melanization assay, and found that Vam6 deficient and overexpression strains display impaired melanization. However, there still appears to be little to no effect on capsule size, despite the statistical significance of the results. The authors still claim that Vam6 is required for capsule formation (line 199: Vam6 is needed for melanin production and capsule formation). A capsule thickness difference in 0.1-0.2 microns is negligible. The Vam6 overexpression strain does appear to show an increase in capsule thickness, but this is not enough to claim that Vam6 is needed for capsule formation.
3) The infection data is convincing, but without an underlying mechanism for Vam6 on the cellular level or the host-pathogen interaction level, the study currently lacks merit.
Reviewer 2 Report
The authors have satisfactorily answered to all my queries
Author Response
We appreciate the reviewer for his/her recognition of our work!